# Schools' Capitalization into Housing Values in a Context of Free School Choices

Mohammad Ismail [ID], Abukar Warsame and Mats Wilhelmsson *[ID]

Division of Real Estate Economics and Finance, Department of Real Estate and Construction Management,
Royal Institute of Technology (KTH), SE-100 44 Stockholm, Sweden; ismail2@kth.se (M.I.);
abukar.warsame@abe.kth.se (A.W.)
* Correspondence: mats.wilhelmsson@abe.kth.se or matswil@kth.se

**Abstract:** The issue of schools and their capitalization in property values has been analyzed extensively. Our contribution is to analyze this effect in an alternative institutional context. In this case study, we analyzed the housing market in Stockholm, Sweden. What distinguishes the Swedish school system is that we have a free choice of schools, which means that a family does not necessarily have to live in a school district to access the schools in that area. This means that families do not have to move to the district to which they intend to send their children but can apply to send them there regardless of whether they live there or not. Nevertheless, families might be interested in living close to good schools to be within walking distance of these schools. This is especially true at the primary school level. Therefore, we analyzed schools' capitalization in property values in the context of free school choice. We used data on transaction prices for condominiums in Stockholm's inner city. The results indicate a capitalization of living close to good schools, but this capitalization is limited. We can show that schools' capitalization depends partly on the quality of the schools and partly on whether or not they are co-located with other externalities, such as green areas. The results also indicate that capitalization is affected by income differences within the city.

**Keywords:** schools; housing values; capitalization

## 1. Introduction

Capitalization theory refers to the fact that property values depend on a community's level of taxation measured by local property tax rates and public services. The quality of schools is an essential local public service and among the most highly valued local amenities [1]. It seems reasonable to expect that the quality of public schools will affect housing prices [1,2].

Based on a simple spatial equilibrium model, Gong and Leung (2020) [3] analyze the implications of education and housing policies in terms of public housing, housing voucher programs, and school finance consolidation in the United States. The results of the policy analyses indicate that public housing policy can induce high sorting among households, which in turn affects the school quality and house prices. Even though housing voucher programs can raise the overall welfare of low-income households, it is the parents who capture the welfare gains more than the children, while public housing programs deliver the opposite result. The well-being of children from low-income households could improve by combining school finance consolidation policy with a public housing program without hurting other households' welfare [3]. In the same context, Leung et al. (2012) [4] assert that the location of public housing units and the number and sizes of units are fundamental policy variables. They indicate the role of public housing and vouchers in improving overall welfare [4]. Hanushek et al. (2011) [5] indicate that private schools provide households freedom to make residential location decisions in relation to the tax–school quality bundles they consume, which alters the spatial distribution of households and subsequently

house prices. Furthermore, private schools contribute to reducing housing and school segregation [5].

The issue of the effect of the quality of schools on property values has been analyzed extensively, but results vary across studies. The variety of results is due to a set of reasons, such as the use of different measures of school quality, study periods, methods used in the estimation, control variables, boundary and neighborhood effects, geographic regions, and education systems—and whether they are free-choice or compulsory school districts [2].

The study's primary purpose is to explore the impact of free-choice schools' quality on surrounding housing values, depending on detailed data for elementary schools, where the elementary school matters more than high school for home buyers or renters, with children driving their preferences for house locations [6–8]. Turnbull and Zheng (2021) [2] indicate that 34 out of 42 papers published after 1980 used elementary school quality measures, where the elementary school measures lead to greater capitalization significance than upper secondary school. Therefore, our hedonic study analyzes elementary school measures using apartment sales in the Stockholm municipality in 2019.

The primary research questions we seek to answer are whether (1) the schools' quality impacts housing values in the context of free school choices, (2) the school capitalization varies by using different measures of school quality, (3) the school capitalization varies with the area income level, and (4) whether other neighborhood amenities affect housing prices.

The research questions are addressed using more than one criterion to classify the elementary school quality in the Stockholm municipality to determine high- and low-quality schools. The hedonic price model is used to examine the relationship between school quality and neighboring housing prices.

The main contribution of our paper is the study of schools' capitalization in housing values regarding the free-choice school system in the Stockholm municipality. We use different criteria to measure the elementary school quality and examine the impact of school quality on apartment prices, not single-family houses—comparing the estimations of capitalization by median income. Moreover, we explore the effect of other neighborhood amenities (green areas and parks) on housing prices to distinguish the school quality effect from other amenities. The present study is the first comprehensive study exploring the effects of school quality on property values in the Stockholm municipality.

The remainder of the paper is divided into six major sections. Section 2 reviews previous studies related to the schools' capitalization in housing values, and we describe the Swedish school system in Section 3. Next, Section 4 describes the data sets and explains our research methodology and the different modeling approaches, whereas Section 5 presents the empirical analysis results. Finally, the paper concludes with a discussion of the study's implications, limitations, and possible routes for future studies in Section 6.

## 2. Literature Review

Researchers have been studying the impact of school quality capitalization on house prices for more than 50 years [2]. Most of the studies were focused on the U.S., where differences in school quality are large, and residential location and school choice can make a big difference in the quality of education and life chances for children [9]. There is something of a consensus in the literature on the importance and role of school quality in the impact and raising capitalized housing values, where the studies consistently found housing valuations to be significantly higher in areas where measured school quality was higher [10]. Researchers used different variables to measure school quality, and they can be categorized into three groups: (1) input-based measures, including per-pupil expenditure, (2) the student/teacher ratio, and (3) all other types of expenditures. Peer effects variables include the percentage of minority students, which reflects the student's racial or ethnic background, and some measures of socioeconomic status. Output-based measures include test score variables [2,11]. Measures based on input have been used by researchers, especially during the early years from 1968 to about 1985, relying on the per-pupil expenditure as a measure of the quality of public schools. The capitalization

results of these studies vary significantly [2], whereas Oates (1969) [12] studied residential communities within fifty-three municipalities in northeastern New Jersey, located within the New York metropolitan region, using expenditure per pupil as a proxy variable for educational services output and a two-stage least-squares estimation technique. The results indicated that the increase in expenditure per pupil from USD 350 to USD 450 pushes house values up by USD 1200, corresponding with other (Sonstelie and Portney, 1980; Bradbury, Case, and Mayer, 2001) results [13,14]. At the same time, Rosen and Fullerton (1977) [15] re-estimated the Oates's study data for the same sample of New Jersey communities to facilitate direct comparisons. The results indicated that expenditure per pupil does not affect house prices in accordance with Brueckner (1979) [16] and Hayes and Taylor (1996) [17].

The second group of school quality measures is peer effects variables. Clapp, Nanda, and Ross (2008) [18] studied the relationship between property values and school district attributes in terms of the racial and ethnic composition of the students and test scores from 1994 to 2004. Increasing segregation in American schools shows that the increased percentage of Hispanics negatively affects housing prices. Moreover, Jud and Watts (1981) [19] found that the racial composition of public schools generally had a negative impact on house prices, and Donald Jud and Watts (1981) [20] indicate that academic quality is more important than the racial ratio of pupils in the schools to determine housing values.

The third group of school quality measures is output-based measures. Brasington (1999) [21] examined thirty-seven school quality measures for six metropolitan areas by running 222 regressions for a traditional least-squares technique, which was repeated for a spatial-autocorrelation estimation technique. The results indicated that the housing market consistently values proficiency test scores. Therefore, these are used to measure school quality. They are the most widely used measure researchers use [22] and the most acceptable variables for measuring school quality [23]. With the increasing availability of school-related variables, most studies indicated the role of school quality measured by test scores in capitalizing on house prices, albeit with various percentages of effect, such as in Black (1999), Downes and Zabel (2002), Rosenthal (2003), Figlio and Lucas (2004), Cheshire and Sheppard (2004), Brasington and Haurin (2006), Bayer, Ferreira, and McMillan (2007), Davidoff and Leigh (2008), Gibbons, Machin, and Silva (2013), La (2015), and Rajapaksa et al. (2020) [24–34].

Many studies used more than one measure for school quality to study school quality impacts on housing values and explore the capitalization significance of those variables. These include studies by Hayes and Taylor (1996), Clark and Herrin (2000), Downes and Zabel (2002), Clapp, Nanda, and Ross (2008), Brasington and Haurin (2006, 2009), Turnbull, Zahirovic-Herbert, and Zheng (2018), and Seo and Simons (2009) [17,18,25,29,35–38].

Most of the research focused on the impact of the school district on housing values, which refers to the school children would attend based on their location of residence [39]. The school district system, which originated in the U.S. and was intended to achieve educational fairness, increased the inequity of educational opportunities. The inequality is caused by the income gap, where wealthy homebuyers are able and willing to pay a premium for a house in a high-quality school district in order to get the advantage of admission opportunities, which increases the impact of the school district and exacerbates the behavior of selecting a school by buying a house. This policy led to an increase in the capitalization effect of primary education [40].

In contrast, the literature has paid relatively little attention to the impact of free-choice schools that admit students regardless of the location of their residence [39]. Few researchers have studied the impact of school choice programs, such as voucher programs, charter schools, magnet schools, and inter-district transfer programs, on property values [41].

Little is known about the effect of free-choice schools on house prices because some economic effects may be positive, and some may be negative. Brehm, Imberman, and Naretta (2017) [42] studied the impact of charter schools, which do not typically have attendance zones and allow students to attend regardless of their location of residence, on housing values in Los Angeles County from 2008 to 2011. The results indicated that, on

average, charter schools had no impact on housing prices. According to Horowitz, Keil, and Spector's (2009) [43] results, elementary charter schools did not affect surrounding property values. Shapiro and Hasset (2013) [44] indicated that opening new charter schools increased the demand for housing in the new schools' respective neighborhoods. Adding one new NYC charter school increased the property values by 3.69%. Walden (1990) [45] studied the effect of elementary magnet schools, which provide parents with the freedom to choose a school in a local school district, and used data gathered in 1987 from Wake County (Raleigh), North Carolina, U.S. The results showed that elementary magnet schools had lower capitalized values and reduced the interdependence of school quality and property values.

Cannon, Danielsen, and Harrison (2015) [46] used a sample of 2933 single-family home transactions and examined the effect of tuition vouchers on home prices in Vermont, U.S., where a tuition voucher program has been in operation since 1869. Tuition vouchers grant Vermont residents living in towns that do not operate public elementary or high schools the right to attend any public school across Vermont. The results found robust evidence that these vouchers increased the values of homes located in jurisdictions that offered school vouchers, and these premiums ranged from 3% to 16%. Merrifield et al. (2011) [47] studied tuition vouchers in San Antonio's Edgewood school district, U.S. The results revealed that the increase in property values in the early years of the program was 9.87%, while it decreased to 1.10% when eligibility requirements were tightened.

Fack and Grenet (2010) [48] studied how private schools affect the capitalization of public-school performance in housing prices—using data on middle schools and housing sales in Paris from 1997 to 2004. The results indicated that increasing public school performance by one standard deviation raises housing prices by 1.4% to 2.4%. Moreover, with the increase in the availability of private schools in the neighborhood, the capitalization of public-school performance in housing prices shrinks by providing an advantageous outside option to parents. Schwartz, Voicu, and Horn (2014) [39] used detailed data for New York City public elementary schools and house sales data from 1988 to 2003. They found that the proximity of alternative school choices reduces the link between zoned schools and property values, and opening a school choice reduces the capitalization from zoned schools into property values by around one-third. Reback (2005) [49] studied the adoption of a public-school choice program and its capitalization effects associated with the diminished importance of school district boundaries, using data from inter-district choice in Minnesota, U.S. The study indicated that property values in districts that accept transfer students decline, while property values increase in districts where students can transfer to preferred school districts.

Among Scandinavian countries, which have similar systems for education and schools, Sweden is distinguished from other Scandinavian countries by its adoption of the free-choice school system. According to the Organization for Economic Co-operation and Development (OECD) Program for International Student Assessment (PISA), Scandinavian schools and pupils are the best-performing worldwide. Harjunen, Kortelainen, and Saarimaa (2018) [9] studied the capitalization role of school quality in house prices in Helsinki. The overall Finish school quality is high and considered the best in the world, and differences, in general, are minor—using data for house prices and test scores for primary schools in Helsinki city, where the city is divided into school catchment areas. By buying a property within the school's catchment area, parents can secure a place for their child in a particular school. The results found that increasing one standard deviation in average test scores increases prices on average by 3%.

Moreover, the price premium is related to pupils' socioeconomic background rather than school effectiveness. Bernelius and Vilkama (2019) [50] pointed out that the residential segregation in Helsinki has grown over the past twenty years, which may also affect the extent of segregation in schools through the composition of pupils. Machin and Salvanes (2016) [51] estimated school quality valuations based on housing prices. They focused on policy reform in Oslo, Norway, in 1997, regarding pupils' choices to attend

high school, where school authorities altered policy from catchment zone enrollment to open enrollment, which allowed pupils to attend any high school. The effects of school performance on housing prices fell by over 50% following the reform.

## 3. The Swedish School System

Sweden underwent a radical transformation in the early 1990s from a central-state governing education to a highly decentralized system promoting school choice and giving parents and children the 'greatest possible freedom' to choose a school. This makes the Swedish educational case unique and prominent in international comparisons. At the same time, other Nordic countries have been much more cautious about adopting a similar shift in education [52]. Sweden has a decentralized education system, where the state steers education goals and learning outcomes through a series of statutes, government orders, and syllabuses, representing a framework for guidelines for all educational aspects. At the same time, the municipalities are responsible for compulsory primary school to upper secondary school and municipal adult education. Municipalities finance the education system by locally collected tax revenues and state grants. There is no fixed salary scale for schoolteachers; teachers' salaries are set individually based on position, skills, qualifications, experience, and performance. Sweden's municipalities have vastly different rates of certified teachers with a high gap between municipalities. The disadvantaged suburbs of major Sweden cities, especially Stockholm, have difficulty attracting skilled teachers to jobs in vulnerable areas, which has an impact on the pupils' outcomes [53–55].

Education in Sweden is free and exempt from fees. Attending primary school is compulsory for nine years. All children aged between seven and sixteen are obliged to attend school. All children shall have equal access to education in Sweden's education system, and the school enrollment is not restricted to the place of residence. The same rules apply to all Swedish school systems—for example, the Swedish Education Act, chapter 1, Section 2 states that "*all children and young people shall, irrespective of gender, place of residence or their social or financial conditions, have equal access to education in the state school system for children and young people. Education in all types of schools shall be of equal value, irrespective of where in Sweden it is provided*" [56,57].

The public school system for children and young people covers preschool, provided by Swedish municipalities for children ages one to five. The Swedish education system is based on comprehensive primary schools with mandatory attendance of nine years for all children aged seven to sixteen. The upper secondary school level or high school consists of study years 10–12, which are optional. There are eighteen regular national programs of three years to choose from, six of which are preparatory for higher education such as university and twelve of which are vocational. Our study covers all the primary schools in the Stockholm municipality, 116 schools divided into two groups, high- and low-quality schools, based on two measures in 2019.

The Swedish school system also includes a growing number of independent schools with public funding, where parents can choose among tuition-free schools, whether municipal or private. Since 1992, the law has provided them with public funding, making them a competitive alternative to municipal schools. In contrast, only a handful of tuition-based private schools are left in Sweden. In Sweden, municipal and independent schools are similar in that they follow the same national curriculum and syllabus. Independent schools must be approved by the Schools Inspectorate. Around 17% of compulsory schools were independent schools, attracting 16% of all compulsory school students from 2020 to 2021 [57–59].

## 4. Data and Methodology

### 4.1. School Quality Measurements

The school data and measures source was the Swedish National Agency for Education, the central administrative authority for the public school system, publicly organized preschool, school-age childcare, and adult education. In order to obtain a more nuanced

picture of schools' grade results than just by publishing the actual grade results that schools achieve, the National Agency for Education developed SALSA (the National Agency for Education's Work Tool for Local Association Analyzes). The analysis tool SALSA presents schools' results of the final grades after some consideration given to the composition of students in order to highlight factors that the school cannot influence but are important to the grade result [56].

Two measures were used to classify the quality of schools. The first measure was the average merit value for a school, which is calculated according to each student's merit value. This measure is the most well-known measure that schools have. It affects the whole group of students, not just those who are close to the approved level, and it is the merit value that students use to apply to upper secondary school. The measure is objectively measured and comparable across schools. A student's merit value is the sum of 17 subjects for students who have studied modern languages as a language choice. Each subject grade is converted thus: E = 10 points, D = 12.5 points, C = 15 points, B = 17.5 points, A = 20 points. Each student's maximum value is 340 points.

Table 1 illustrates that the average merit value for all primary schools in Stockholm municipality is 252. For high-quality schools, the merit value is 284, around 32 points over the average. At the same time, the average merit value for low-quality schools is 218, 34 points lower than the average. The difference between the average high- and low-quality schools is sixty-six points. At the same time, the difference between the minimum value of high- and low-quality schools is 117 points, and the maximum value is 67 points, which shows the big gap in schools' result outcomes.

**Table 1.** Descriptive statistics for Measure 1: merit value.

| Schools | Obs | Mean | Std. Dev. | Min | Max |
|---|---|---|---|---|---|
| All Schools | 116 | 252.474 | 30.881 | 153 | 308 |
| High-Quality Schools | 40 | 283.575 | 11.989 | 270 | 308 |
| Low-Quality Schools | 40 | 217.525 | 19.177 | 153 | 241 |

The second measure was the expected average merit value for schools in relation to the students' background factors. The measure is calculated by the National Agency for Education and is based on a regression model that considers certain background factors. The background factors consist of the parents' average level of education, the proportion of new immigrants, and the proportion of boys. The parents' average level of education is calculated by converting the completed primary school to one point, the completed upper secondary education to two points, and post-secondary education to three points. The average level of education is calculated for each school unit. The proportion of new immigrants refers to the proportion of students who have been registered in Sweden for the first time in the last four years. The proportion of boys is also calculated for each school unit. A high average level of education among parents gives a higher model-calculated expected average merit value to the school. A high proportion of boys and newly immigrated students are expected to lower the average merit value of the school.

Table 2 shows that the parents' average level of education in high-quality schools is 2.7, which is higher than the average of all schools at 2.4, whereas for low-quality schools, the parents' average level is 2. The result indicates a difference in socioeconomic factors for the students' parents. The percentage of students with foreign backgrounds enrolled in low-quality schools is around ten times that of high-quality schools, and this reflects schools' segregation based on racial and ethnic factors. The percentage of boys is higher in low-quality schools at 50% compared to that of high-quality schools at 47%. The expected average value for all schools is 239, while for high-quality schools, it is 263. For low-quality schools, the expected average value is 209, with the gap between high- and low-quality schools being 54 points.

**Table 2.** Descriptive statistics for Measure 2: expected average value.

| All Schools | Obs | Mean | Std. Dev. | Min | Max |
|---|---|---|---|---|---|
| Parents' average level of education | 116 | 2.424 | 0.314 | 1.4 | 2.9 |
| The proportion of new immigrants | 116 | 4.871 | 5.894 | 0 | 26 |
| Percentage of boys | 116 | 49.767 | 10.627 | 17 | 84 |
| Expected average value | 116 | 239.052 | 26.126 | 170 | 276 |
| High-Quality Schools | | | | | |
| Parents' average level of education | 40 | 2.705 | 0.093 | 2.5 | 2.9 |
| The proportion of new immigrants | 40 | 1.15 | 1.748 | 0 | 7 |
| Percentage of boys | 40 | 47.4 | 10.412 | 17 | 71 |
| Expected average value | 40 | 263.35 | 6.754 | 252 | 276 |
| Low-Quality Schools | | | | | |
| Parents' average level of education | 40 | 2.067 | 0.239 | 1.4 | 2.4 |
| The proportion of new immigrants | 40 | 9.725 | 6.695 | 0 | 26 |
| Percentage of boys | 40 | 50.65 | 10.895 | 21 | 72 |
| Expected average value | 40 | 208.775 | 18.874 | 170 | 235 |

The following figures (Figures 1 and 2) show the distribution of high- and low-quality schools in the Stockholm municipality based on the two measures used in the study. We observe a similarity in the distribution in the two figures, as most of the high-quality schools are concentrated in central Stockholm, characterized by a high level of income and a low percentage of non-Swedes. Low-quality schools are concentrated in the suburbs of Stockholm, where they are characterized by a low level of income and a high proportion of non-Swedes. This concentration in the distribution is evident in Figure 2, which considers the background factors of the students. This result indicates high segregation in Stockholm schools, thereby reflecting residential segregation based on socioeconomic and racial factors.

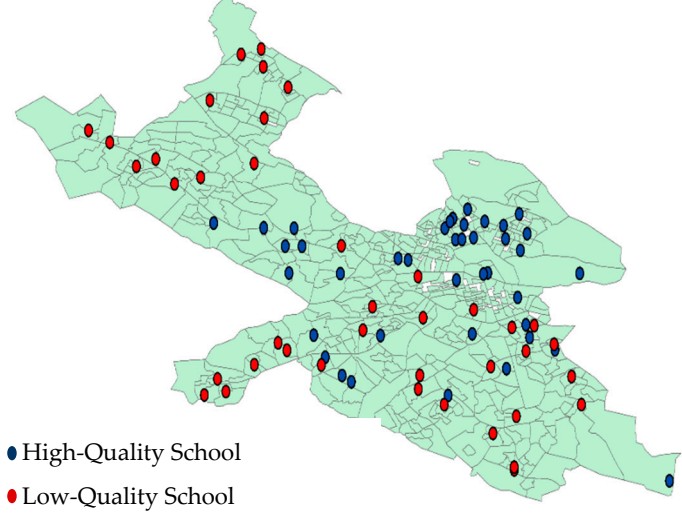

● High-Quality School

● Low-Quality School

**Figure 1.** High- and low-quality schools based on Measure 1.

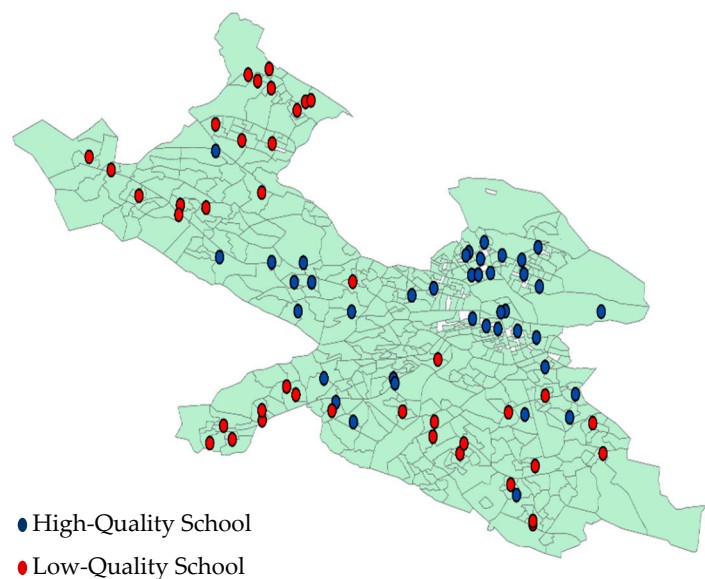

● High-Quality School

● Low-Quality School

**Figure 2.** High and low-quality schools based on Measure 2.

*4.2. Housing Transactions*

Data on housing values were taken from residential property transactions of owner-occupied condominium apartments for 2019. The case study used was the municipality of Stockholm, the capital of Sweden. Table 3 presents the descriptive statistics and illustrates the characteristics of the transactions. Svensk Mäklarstatistik provided the transaction data for the study.

**Table 3.** Descriptive statistics of the transactions.

| Variable | Mean | Std. Dev. | Min | Max |
|---|---|---|---|---|
| Price | 3,834,265.5 | 1,782,632.2 | 1,605,000 | 12,050,000 |
| Living Area | 61.531 | 23.269 | 22.3 | 142 |
| Number of rooms | 2.443 | 1.047 | 1 | 9 |
| Monthly Fee | 3304.9 | 1229.932 | 660 | 6935 |
| Build year | 1960.14 | 39.59 | 1500 | 2022 |
| Apartment floor | 2.27 | 2.594 | −3 | 85 |
| New | 0.028 | 0.165 | 0 | 1 |
| Elevator | 0.582 | 0.493 | 0 | 1 |
| Distance to subway | 0.586 | 0.439 | 0.001 | 4.693 |
| Distance to a shopping mall | 1.645 | 1.009 | 0.008 | 4.939 |
| Distance to CBD | 5.309 | 3.274 | 0.172 | 16.474 |

Notes: The number of observations is 12,807; prices are in Swedish Krona (SEK), and distance is in km.

Table 3 presents descriptive statistics regarding the data on condominium sales. The observed number of apartments amounts to 12,807. The average price is approximately SEK 4 million, with a standard deviation of SEK 1.8 million. The average size of an apartment living area is 62 square meters, with a standard deviation of 23 square meters. The average monthly fee paid by apartment owners to the building association for maintenance and upkeep is SEK 3300, with a standard deviation of SEK 1200. The average number of apartment rooms equals two, and the average number of building floors equals two. The average build year is 1960, with a standard deviation of 40 years. The data also include binary variables of whether the building is new or has an elevator (1 if it exists and 0 elsewhere). The average distance to the subway is around 0.6 km, and the average distance to the shopping mall is 1.6 km. In comparison, the average distance to the Central Business District (CBD) is around five kilometers, which means that many apartments do not have a location close to the city center.

### 4.3. Methodology

To address the main research questions in the study, we used the hedonic price model to estimate how proximity to school affects housing prices. The hedonic method was developed and presented in a seminal article by Rosen (1974) [60]. The model is used widely in school capitalization studies and is the most used methodology for exploring the relationships between school networks and property values [22,61].

The hedonic model is based on housing as a heterogeneous commodity with a bundle of varied structure, neighborhood, and property location characteristics. Structure characteristics include the property size, apartment floor, and building age. Traffic noise, crime rate, and air pollution are examples of neighborhood attributes. The property location includes the distance to schools, shops, the CBD, and the subway. All these attributes contribute to determining the house price [22].

School quality is a neighborhood amenity. Our main task was to separate school quality from other neighborhood effects. One way was to include as many relevant neighborhood characteristics as possible to control non-school neighborhood effects. Including more neighborhood characteristics and variables reduced the significance of school quality capitalization, where the additional variables controlled for correlated effects unrelated to school quality. We also included the binary variable for a park located on the school borders in the hedonic models to differentiate school quality effects on house values from other neighborhood amenity impacts.

Income level is often used in school quality capitalization studies [2]. We did not include it as a control variable. Instead, we tested spatial parameter heterogeneity by estimating different models in high-income and low-income areas.

The present study estimated the hedonic price model based on property attributes that significantly affect housing prices. The focus was on the attribute proximity to schools. The model that was estimated can be illustrated in the following equation:

$$Ln(P_i) = \alpha + \beta_1 Ln(X_i) + \beta_2 Ln(S_i) + \beta_3 Ln(D_i) + \varepsilon_i$$

where $Ln(P_i)$ is the logarithm of the transaction price of the property, $i$, and $Ln(X_i)$ is a logarithm of housing and neighborhood characteristics. $Ln(S_i)$ is the logarithm of distance or proximity to the school, and $Ln(D_i)$ is a logarithm of binary fixed effects. $\alpha$ represents the constant term, and $\beta_1$, $\beta_2$, and $\beta_3$ are the coefficients of the variables to be estimated. Finally, $\varepsilon_i$ is the random error term of the model.

Ordinary least squares (OLS) was used. We tested the functional form with a Box-Cox transformation of the dependent and independent variables as the linear Box-Cox function seems to be the functional form of choice [62].

A key assumption when we estimated the above models with ordinary least squares was the question of whether the independent variables were exogenously given. Endogeneity can be caused by omitted variables, measurement errors, and reverse causality [63]. We controlled the former cause by including all relevant housing-value-affecting attributes and fixed geographical and monthly effects. We checked measurement errors by analyzing included variables and trying to identify potential measurement errors. Reverse causality is, of course, the most challenging cause to control. A common approach to controlling for reverse causality is to use instrument variables [64]. This means we want to identify variables correlated with the endogenous variable but not with the error term. It is notoriously difficult to find suitable instrument variables that are both valid and strong that are strictly exogenous [65]. The use of time-lagged instrument variables is relatively common in empirical research (recently in Dong et al., 2018, and Ujah et al., 2020) [66,67]. However, as Reed (2015) has shown, using time-lagged instrument variables is not without problems, and we may still have simultaneity issues and problems testing hypotheses. We used the instrument variable approach with time-lagged instrument variables as a robustness test of our default models estimated using ordinary least squares [68].

## 5. The Econometric Analysis

In this section, we discuss the results of the econometric analysis by estimating the hedonic equation based on two measures for school quality that were used in the study. The analysis took place in four steps: In the first step, we estimated a model where we related housing apartment prices to the usual value-affecting attributes but also to the distance to the nearest school (divided into high- and low-quality schools based on the National Agency for Education Quality Measure 1, average merit value). Of course, it may be that schools are co-located with many other value-influencing attributes in the residential area, such as parks and green areas. Older schoolyards are also a green area in the residential area's micro-location. Therefore, in the next step, we estimated a model where, with the help of Google Maps and Google Street View, we created a variable that indicates whether a school is close to or adjacent to a green area. The hypothesis we wanted to test was whether proximity to a green area or a school raises apartment values. In the third step, we analyzed whether the income level in the residential area impacts the capitalization of proximity to a school. Finally, we also estimated all models, but with School Quality Measure 2, which was also calculated by the National Agency for Education and was based on the expected average value. All continuous variables were transformed with the natural logarithm (based on the Box-Cox test), and thus, the interpretation of the parameter estimates (coefficients) can be expressed in elasticities.

### 5.1. Step 1: Default Model (Based on Measure 1 for School Quality)

We estimated the hedonic price model (HPM) using the Box-Cox transformation to find the empirically best form of the hedonic function. The logarithm specification form of the hedonic pricing model was preferred, whereas the model with a logarithmic form had high explanatory power. Therefore, the logged dependent variable (housing price) was regressed as a function of a set of logged attributes and unlogged attributes.

Table 4 shows the results of the default OLS model, relying on Measure 1 for school quality. The estimated hedonic model provided a high goodness-of-fit. We controlled spatial location by including zip codes and coordinates in the model. Additionally, the monthly fixed effect was included to capture possible housing price changes within the study year. The model explains around 87% of the total variation in housing prices. The results suggest that most of the property attributes (independent variables) incorporated in the model were simultaneously highly significant and fit well into the model.

The results showed positive coefficients and are highly significant (living area, number of rooms, apartment floor, and new building). Hence, these property attributes are positively related to the housing apartment price. Increasing the living area by 1% increases the price by about 0.73%. Buyers are willing to pay a premium to live in a larger housing unit—the positive and larger coefficient indicate this. In line with this, increasing the number of rooms by 1% increases the price by about 0.08%. Increasing the apartment floor by 1% increases the price by 0.04%. People usually prefer to live on a higher floor, to get a better view, have cleaner air, and experience less noise. The house price increases by 0.19% if the property is new. The results show that if the building includes an elevator, the positive effect is less than 0.01%, and it is not significant in all models.

The results show negative coefficients and are highly significant (monthly fee, dist_subway, and dist_CBD). For example, if the monthly fee increases by 1%, the apartment price decreases by around 0.1%. When increasing the distance between a property and the subway station by 1%, the apartment price drops by 0.01% in Model 2 and 4 and around 0.02% in Model 1 and 3. Increasing the distance by 1% between a property and the CBD reduces the apartment price by more than 0.2%, while the distance to the shopping mall shows a significant and low coefficient of around 0.01% in the model (2).

Regarding the accessibility of primary schools, the results suggest that an increase in the distance of a property to a primary school by 1% will reduce the property price by 0.021%. Increasing the distance of a property to a high-quality school by 1% will reduce apartment prices by 0.0432%. This indicates that homebuyers are willing to pay a certain

premium to live near a primary school, and this premium increases for houses near a high-quality school. In comparison, the increase in the distance of a property to a low-quality school by 1% increased the property price by 0.0072%. These results indicate the high capitalization role of high-quality schools in housing prices.

**Table 4.** HPM results for Measure 1.

| Variable | Model 1 | Model 2 | Model 3 | Model 4 |
|---|---|---|---|---|
| Ln (Living area) | 0.729 *** | 0.733 *** | 0.728 *** | 0.731 *** |
|  | (77.14) | (78.32) | (76.77) | (77.26) |
| Ln (number of rooms) | 0.0818 *** | 0.0775 *** | 0.0825 *** | 0.0802 *** |
|  | (10.86) | (10.38) | (10.91) | (10.64) |
| Ln (monthly fee) | −0.104 *** | −0.100 *** | −0.103 *** | −0.105 *** |
|  | (−17.37) | (−16.84) | (−17.07) | (−17.55) |
| Ln (build year) | 0.165 | 0.202 ** | 0.0966 | 0.263 ** |
|  | (1.629) | (2.018) | (0.952) | (2.549) |
| Ln (apartment floor) | 0.0402 *** | 0.0400 *** | 0.0400 *** | 0.0403 *** |
|  | (17.15) | (17.21) | (16.98) | (17.16) |
| New | 0.186 *** | 0.181 *** | 0.187 *** | 0.185 *** |
|  | (15.27) | (15.00) | (15.32) | (15.20) |
| Elevator | 0.00343 | 0.00726 ** | 0.00299 | 0.00416 |
|  | (0.996) | (2.124) | (0.866) | (1.208) |
| Ln (dist_subway) | −0.0168 *** | −0.00888 *** | −0.0213 *** | −0.0121 *** |
|  | (−7.203) | (−3.744) | (−9.122) | (−4.854) |
| Ln (dist_shop) | 0.00440 | −0.0118 *** | 0.00415 | 0.00416 |
|  | (1.357) | (−3.502) | (1.267) | (1.281) |
| Ln (dist_CBD) | −0.235 *** | −0.218 *** | −0.242 *** | −0.224 *** |
|  | (−32.06) | (−29.59) | (−33.24) | (−29.44) |
| Ln (N_DS_All_S) | −0.0210 *** |  |  | −0.0474 *** |
|  | (−8.500) |  |  | (−8.145) |
| Ln (N_DS_G_S) |  | −0.0432 *** |  |  |
|  |  | (−16.35) |  |  |
| Ln (N_DS_B_S) |  |  | 0.00718 ** |  |
|  |  |  | (2.349) |  |
| Constant | 11.98 *** | 11.78 *** | 12.31 *** | 11.41 *** |
|  | (15.87) | (15.78) | (16.24) | (14.93) |
| Observations | 9446 | 9446 | 9446 | 9446 |
| R-squared | 0.872 | 0.874 | 0.871 | 0.871 |

Notes: 1—t-statistics in parentheses *** $p < 0.01$, ** $p < 0.05$; 2—LnN_DS_All_S = Log (near distance to all schools), LnN_DS_G_S = Log (near distance to high-quality schools), InN_DS_B_S0 = Log (near distance to low-quality schools).

If we analyze Model 4 (instrument variable approach) in Table 4, we find that proximity to schools has expected signs of effect as in Models 1–3. It is worth noting that the effect is twice as large as in Model 1, which is a comparable model. However, caution in interpretation should be exercised, as time-lagged variables may be invalid. A cautious interpretation is that there is a positive effect on housing prices of living near a school. Furthermore, it is not a question of reverse causality, as we also note the same empirical results when using 20-year time-lagged instrument variables. The lack of instrument variables for distances to schools of better or worse quality means that we interpreted the empirical analysis without using instrument variables.

### 5.2. Step 2: Micro-location—Green Areas

Table 5 illustrates the results of the hedonic model, including an indicator of if a school is located in the vicinity of green areas. The accessibility of primary schools indicates that increasing the distance of a property to a high-quality school close to a park by 1% will reduce property prices by 0.0482%, while increasing the distance of a property to a high-quality school not close to a park by 1% will reduce property prices by 0.0345%. This indicates that homebuyers are willing to pay a certain premium to live near a high-quality school, and this premium increases for schools close to a park or green area.

The proximity to a low-quality school indicates that increasing the distance of a property to a low-quality school close to a park by 1% will increase property prices by 0.026%, while for a low-quality school not close to a park, house prices increase by 0.0066%. The results indicate that the proximity to low-quality schools has a negative effect on housing prices, even though the properties' locations can explain the closeness to parks.

**Table 5.** HPM results are controlled by being close to a park for Measure 1.

| Variable | G1 (1) | G0 (2) | B1 (3) | B0 (4) |
|---|---|---|---|---|
| Ln (Living area) | 0.725 *** | 0.737 *** | 0.729 *** | 0.728 *** |
| | (77.39) | (78.17) | (77.10) | (76.75) |
| Ln (number of rooms) | 0.0808 *** | 0.0765 *** | 0.0808 *** | 0.0829 *** |
| | (10.83) | (10.19) | (10.72) | (10.96) |
| Ln (monthly fee) | −0.101 *** | −0.102 *** | −0.101 *** | −0.104 *** |
| | (−16.94) | (−17.16) | (−16.85) | (−17.19) |
| Ln (build year) | 0.0102 | 0.205 ** | 0.0625 | 0.0952 |
| | (0.101) | (2.031) | (0.619) | (0.938) |
| Ln (apartment floor) | 0.0416 *** | 0.0394 *** | 0.0402 *** | 0.0400 *** |
| | (17.87) | (16.88) | (17.12) | (16.99) |
| New | 0.178 *** | 0.185 *** | 0.183 *** | 0.187 *** |
| | (14.77) | (15.31) | (15.01) | (15.34) |
| Elevator | 0.00804 ** | 0.00480 | 0.00409 | 0.00276 |
| | (2.344) | (1.399) | (1.188) | (0.797) |
| Ln (dist_subway) | −0.00978 *** | −0.0142 *** | −0.0214 *** | −0.0211 *** |
| | (−4.128) | (−6.094) | (−9.329) | (−9.050) |
| Ln (dist_shop) | −0.00848 ** | −0.00745 ** | 0.00183 | 0.00466 |
| | (−2.546) | (−2.211) | (0.560) | (1.431) |
| Ln (dist_CBD) | −0.215 *** | −0.217 *** | −0.239 *** | −0.242 *** |
| | (−28.98) | (−28.86) | (−32.91) | (−33.15) |
| Ln (N_DS_G1_S) | −0.0482 *** | | | |
| | (−15.52) | | | |
| Ln (N_DS_G0_S) | | −0.0345 *** | | |
| | | (−12.88) | | |
| Ln (N_DS_B1_S) | | | 0.0255 *** | |
| | | | (8.013) | |
| Ln (N_DS_B0_S) | | | | 0.00659 ** |
| | | | | (1.982) |
| Constant | 13.34 *** | 11.72 *** | 12.40 *** | 12.33 *** |
| | (17.83) | (15.60) | (16.47) | (16.26) |
| Observations | 9446 | 9446 | 9446 | 9446 |
| R-squared | 0.874 | 0.873 | 0.871 | 0.871 |

Note: 1—t-statistics in parentheses *** $p < 0.01$, ** $p < 0.05$; 2—G1: high-quality school close to park, G0: high-quality school not close to park, B1: low-quality school close to park, B0: low-quality school not close to park.

### 5.3. Step 3: Spatial Parameter Heterogeneity

Table 6 illustrates the results of the hedonic model for high- and low-income areas. The results regarding the accessibility of primary schools indicate that there is no significant effect on the distance of a property in a high-income area to a primary school. Increasing a property's distance from a low-income area to a primary school by 1% will reduce property prices by 0.0417%. This indicates that low-income homebuyers are willing to pay a certain premium to live near a primary school, unlike homebuyers in a high-income area.

The proximity to a high-quality school indicates that increasing the distance between a property and a high-quality school in a high-income area by 1% will reduce property prices by 0.009%. In comparison, proximity reduces property prices by 0.0625% in low-income areas if the distance increases by 1%. This means that people in low-income areas are willing to pay more for apartments near a high-quality school.

The proximity to a low-quality school indicates that the increase in a property's distance from a high-income area to a low-quality school by 1% will increase property prices by 0.0176%. There is no significant effect on property proximity in a low-income area to a low-quality school.

Furthermore, we used polynomial regression, described in Peixoto (1990) [69], as a robustness test for potential non-linearity between the dependent variable housing price and the independent variable distance to schools. By estimating a polynomial model, we could graphically illustrate the relationship, as shown in Figure 3. Notable from the polynomial regression is that the capitalization effect is positive and very local. From about 300 m from the school, the capitalization effect almost disappears.

**Table 6.** HPM results are controlled by income for Measure 1.

| Variable | High Inc (1) | Low Inc (2) | High Inc (3) | Low Inc (4) | High Inc (5) | Low Inc (6) |
|---|---|---|---|---|---|---|
| Ln (Living area) | 0.780 *** | 0.615 *** | 0.780 *** | 0.627 *** | 0.779 *** | 0.615 *** |
| | (76.52) | (38.03) | (76.57) | (39.20) | (76.58) | (37.46) |
| Ln (number of rooms) | 0.0863 *** | 0.122 *** | 0.0859 *** | 0.113 *** | 0.0865 *** | 0.124 *** |
| | (10.11) | (10.41) | (10.07) | (9.780) | (10.16) | (10.40) |
| Ln (monthly fee) | −0.0664 *** | −0.196 *** | −0.0656 *** | −0.189 *** | −0.0648 *** | −0.197 *** |
| | (−10.96) | (−16.58) | (−10.83) | (−16.16) | (−10.70) | (−16.39) |
| Ln (build year) | −1.022 *** | 1.561 *** | −0.997 *** | 1.550 *** | −0.989 *** | 1.435 *** |
| | (−9.647) | (8.154) | (−9.420) | (8.211) | (−9.368) | (7.396) |
| Ln (apartment floor) | 0.0407 *** | 0.0356 *** | 0.0407 *** | 0.0348 *** | 0.0405 *** | 0.0342 *** |
| | (15.28) | (9.813) | (15.29) | (9.722) | (15.25) | (9.291) |
| New | 0.102 *** | 0.209 *** | 0.101 *** | 0.203 *** | 0.106 *** | 0.203 *** |
| | (6.603) | (12.37) | (6.528) | (12.19) | (6.898) | (11.83) |
| Elevator | 0.0185 *** | −0.0121 ** | 0.0189 *** | −0.00403 | 0.0177 *** | −0.0118 ** |
| | (4.512) | (−2.379) | (4.613) | (−0.797) | (4.313) | (−2.279) |
| Ln (dist_subwa) | −0.0150 *** | −0.0266 *** | −0.0127 *** | −0.0179 *** | −0.0167 *** | −0.0351 *** |
| | (−5.180) | (−7.796) | (−4.266) | (−5.144) | (−5.790) | (−10.10) |
| Ln (dist_shop) | −4.28 × $10^{-5}$ | 0.0137 *** | −0.00413 | −0.00427 | −0.00114 | 0.0141 *** |
| | (−0.0110) | (2.771) | (−0.996) | (−0.847) | (−0.293) | (2.802) |
| Ln (dist_CBD) | −0.166 *** | −0.312 *** | −0.164 *** | −0.265 *** | −0.167 *** | −0.339 *** |
| | (−20.99) | (−22.73) | (−20.72) | (−18.72) | (−21.21) | (−24.68) |
| Ln (N_DS_All_S) | −0.00196 | −0.0417 *** | | | | |
| | (−0.676) | (−10.93) | | | | |
| Ln (N_DS_G_S) | | | −0.00934 *** | −0.0625 *** | | |
| | | | (−2.959) | (−15.03) | | |
| Ln (N_DS_B_S) | | | | | 0.0176 *** | −0.00329 |
| | | | | | (4.325) | (−0.787) |
| Constant | 20.29 *** | 2.804 ** | 20.13 *** | 2.898 ** | 19.88 *** | 3.524 ** |
| | (25.71) | (1.966) | (25.54) | (2.059) | (25.11) | (2.438) |
| Observations | 5279 | 4167 | 5279 | 4167 | 5279 | 4167 |
| R-squared | 0.897 | 0.807 | 0.897 | 0.812 | 0.897 | 0.802 |

t-statistics in parentheses *** $p < 0.01$, ** $p < 0.05$.

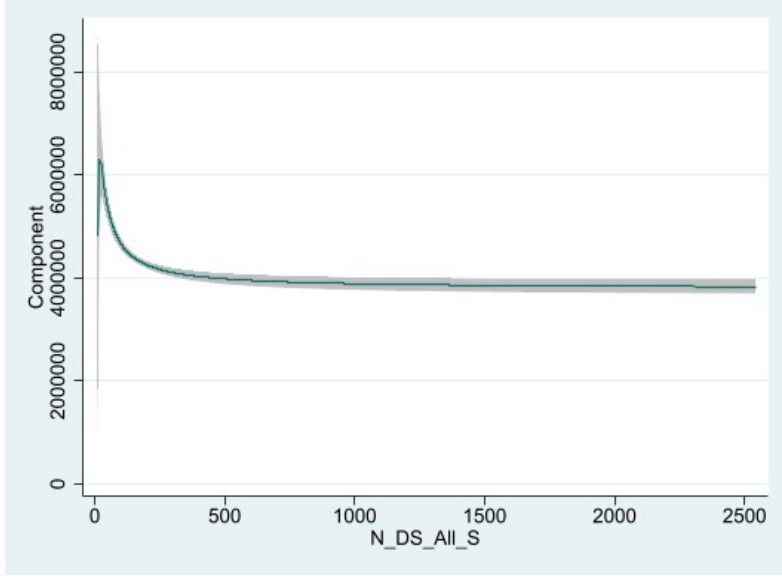

**Figure 3.** Polynomial plotting.

### 5.4. Step 4: School Quality Based on Expected Merit Value (Measure 2)

We estimated the hedonic price model for school quality based on expected merit value (Measure 2) and applied the same methodology we used for Measure 1. Tables A1–A3 (in the Appendix A) show the results of the default OLS model, relying on Measure 2 for school quality. We can note that the goodness-of-fit is high at around 87% in the models. All estimated parameters have expected signs and reasonable magnitude, where the property attributes and results are close to the results based on Measure 1. We analyze the results of the proximity of a property to schools, explore this effect, and investigate the capitalization role of schools on housing prices.

The results in Table A1 (in the Appendix A) suggest that a 1% increase in the distance of a property from a high-quality school reduces the price of a property by 0.0392%, which is less than the first measure. In comparison, increasing the distance of a property from a low-quality school by 1% increases the price of the property by 0.00704%. This means that being near a high-quality school increases house prices, while being near a low-quality school decreases house prices. This emphasizes the capitalization role of the quality of schools on housing prices in the context of free school choices.

The results in Table A2 (in the Appendix A) indicate that increasing the distance of a house in a high-income area from a high-quality school by 1% reduces the price of the house by 0.011%, while a similar increase in a low-income area reduces the property price by 0.065%. The results suggest that a 1% increase in the distance of a property in a high-income area from a low-quality school will increase the price of the property by 0.0124%. At the same time, there is no significant effect for the proximity of houses to low-quality schools in low-income areas. These results suggest that households in low-income areas appreciate being near to schools, especially high-quality schools, more than households in high-income areas.

Finally, the results in Table A3 (in the Appendix A) suggest that homebuyers are willing to pay a premium to live near a high-quality school, especially for a house close to a park, where increasing the distance of a property to a high-quality school close to a park by 1% reduces the price of a property by 0.043%, and a similar move from a high-quality school not close to a park increases the price by 0.0313%. Regarding low-quality schools, if there is a 1% increase in the distance between a property and a low-quality school that is close to a park, the property's price will increase by 0.0394%. Conversely, if there is a 1% increase in the distance between a property and a low-quality school that is not close to a park, the property's price will decrease by 0.0111%.

## 6. Conclusions

Most research studies on property values in relation to school performance often focus on the impact of the quality of zoned schools on housing values. Zoned schools are schools that children would attend based on their address locations. The possible impact of free-choice schools that admit students without geographic criteria constraints has not been extensively explored. In this paper, we tried to quantify whether the proximity to schools is capitalized on the prices of apartments within the context of free-choice schools.

We also extended our estimation of this capitalization based on the income levels of households and considered using different school quality measures. The two school quality measures considered in this study were expected to produce different quality measurements since one of them implied distinct sociodemographic factors, such as parents' education and the number of new immigrants in these schools. Compared to low-quality schools, in high-quality schools, there are fewer new immigrants, and parents have higher levels of education. However, the estimated results seemed not to differ between the two school quality measurements.

School proximity is capitalized on property values when parents have the freedom to choose the school's system. Our results indicate that this capitalization is more substantial when properties are close to high-quality schools than when properties are close to low-quality schools. However, when income is considered, the capitalization of school quality on apartment prices produces mixed results where low-income households' closeness to good schools has the highest capitalization value.

One of the plausible policy implications of this study is how programs' intentions to reduce segregation might be hampered by the locations of high- or low-quality schools in relation to the income levels of households. If good/bad schools are concentrated in certain areas, it will increase residential segregation in the city since low-income households have fewer mobility options than high-income households. If school quality becomes highly relevant in households' location decisions, the result could be the Tiebout model of household sorting, where high-income households will choose to co-locate with similar

high-income households and close to good-quality schools. This might also lead to the displacement of low-income households through the house-price mechanism, forcing them to move closer to bad-quality schools. This study shows intertwined relationships between residential segregation, income, and school quality. Policies or programs intended to address segregation concerns should consider the location of schools and their performance.

Our study did not differentiate between municipal and private schools. Currently, both types of schools receive the same amount of funding per child from the government. However, a current proposal would increase municipal school funding at the expense of privately administered schools. Further research on the impact of school quality on housing values from municipal and private school segmentation perspectives and a possible effect of such a proposal on school quality would be needed.

**Author Contributions:** Conceptualization, M.W.; methodology, analysis, initial draft preparation, M.I., A.W. and M.W.; writing—review and editing, M.I., A.W. and M.W. All authors have read and agreed to the published version of the manuscript.

**Funding:** This research was funded by Housing 2.0 (Bostad 2.0).

**Institutional Review Board Statement:** Not applicable.

**Informed Consent Statement:** Not applicable.

**Data Availability Statement:** The transaction data were provided by Svensk Mäklarstatistik https://www.maklarstatistik.se/ (accessed on 1 March 2021). School quality data are available at: https://www.skolverket.se/ (accessed on 1 January 2022).

**Acknowledgments:** We thank the research project Housing 2.0 (Bostad 2.0) for financial support.

**Conflicts of Interest:** The authors declare no conflict of interest.

## Appendix A

**Table A1.** HPM results for Measure 2.

| Variable | Model 1 | Model 2 | Model 3 |
|---|---|---|---|
| Ln (living area) | 0.729 *** | 0.725 *** | 0.729 *** |
|  | (77.14) | (77.39) | (76.81) |
| Ln (number of rooms) | 0.0818 *** | 0.0812 *** | 0.0822 *** |
|  | (10.86) | (10.88) | (10.87) |
| Ln (monthly fee) | −0.104 *** | −0.0984 *** | −0.103 *** |
|  | (−17.37) | (−16.51) | (−17.13) |
| Ln (build year) | 0.165 | 0.213 ** | 0.107 |
|  | (1.629) | (2.119) | (1.048) |
| Ln (apartment floor) | 0.0402 *** | 0.0405 *** | 0.0401 *** |
|  | (17.15) | (17.39) | (17.03) |
| New | 0.186 *** | 0.175 *** | 0.186 *** |
|  | (15.27) | (14.50) | (15.26) |
| Elevator | 0.00343 | 0.00562 | 0.00299 |
|  | (0.996) | (1.644) | (0.865) |
| Ln (dist_subway) | −0.0168 *** | −0.00882 *** | −0.0212 *** |
|  | (−7.203) | (−3.689) | (−9.099) |
| Ln (dist_shop) | 0.00440 | −0.00855 ** | 0.00451 |
|  | (1.357) | (−2.566) | (1.384) |
| Ln (dist_CBD) | −0.235 *** | −0.230 *** | −0.242 *** |
|  | (−32.06) | (−31.72) | (−33.23) |
| Ln (N_DS_All_S) | −0.0210 *** |  |  |
|  | (−8.500) |  |  |
| Ln (N_DS_G_S) |  | −0.0392 *** |  |
|  |  | (−15.46) |  |
| Ln (N_DS_B_S) |  |  | 0.00704 ** |
|  |  |  | (2.295) |
| Constant | 11.98 *** | 11.69 *** | 12.23 *** |
|  | (15.87) | (15.63) | (16.06) |
| Observations | 9446 | 9446 | 9446 |
| R-squared | 0.872 | 0.874 | 0.871 |

Notes: 1—t-statistics in parentheses *** $p < 0.01$, ** $p < 0.05$; 2—InN_DS_All_S = Log (near distance to all schools), InN_DS_G_S = Log (near distance to high-quality schools), InN_DS_B_S0 = Log (near distance to low-quality schools).

**Table A2.** HPM results are controlled by income for Measure 2.

| Variable | High Inc (1) | Low Inc (2) | High Inc (3) | Low Inc (4) | High Inc (5) | Low Inc (6) |
|---|---|---|---|---|---|---|
| Ln (living area) | 0.780 ** | 0.615 *** | 0.779 *** | 0.614 *** | 0.780 *** | 0.615 *** |
| | (76.52) | (38.03) | (76.40) | (38.66) | (76.61) | (37.43) |
| Ln (number of rooms) | 0.0863 *** | 0.122 *** | 0.0865 *** | 0.117 *** | 0.0861 *** | 0.124 *** |
| | (10.11) | (10.41) | (10.15) | (10.23) | (10.10) | (10.41) |
| Ln (monthly fee) | −0.0664 *** | −0.196 *** | −0.0651 *** | −0.184 *** | −0.0658 *** | −0.197 *** |
| | (−10.96) | (−16.58) | (−10.75) | (−15.84) | (−10.87) | (−16.39) |
| Ln (build year) | −1.022 *** | 1.561 *** | −0.978 *** | 1.383 *** | −0.982 *** | 1.429 *** |
| | (−9.647) | (8.154) | (−9.213) | (7.380) | (−9.229) | (7.375) |
| Ln (apartment floor) | 0.0407 *** | 0.0356 *** | 0.0407 *** | 0.0365 *** | 0.0409 *** | 0.0342 *** |
| | (15.28) | (9.813) | (15.30) | (10.26) | (15.38) | (9.294) |
| New | 0.102 *** | 0.209 *** | 0.0990 *** | 0.191 *** | 0.103 *** | 0.203 *** |
| | (6.603) | (12.37) | (6.414) | (11.50) | (6.678) | (11.84) |
| Elevator | 0.0185 *** | −0.0121 ** | 0.0189 *** | −0.00608 | 0.0179 *** | −0.0118 ** |
| | (4.512) | (−2.379) | (4.610) | (−1.213) | (4.367) | (−2.276) |
| Ln (dist_subwa) | −0.0150 *** | −0.0266 *** | −0.0119 *** | −0.0170 *** | −0.0157 *** | −0.0350 *** |
| | (−5.180) | (−7.796) | (−3.984) | (−4.951) | (−5.464) | (−10.03) |
| Ln (dist_shop) | $-4.28 \times 10^{-5}$ | 0.0137 *** | −0.00374 | −0.00884 * | −0.000607 | 0.0140 *** |
| | (−0.0110) | (2.771) | (−0.929) | (−1.754) | (−0.156) | (2.780) |
| Ln (dist_CBD) | −0.166 *** | −0.312 *** | −0.165 *** | −0.303 *** | −0.167 *** | −0.339 *** |
| | (−20.99) | (−22.73) | (−20.89) | (−22.62) | (−21.25) | (−24.65) |
| Ln (N_DS_All_S) | −0.00196 | −0.0417 *** | | | | |
| | (−0.676) | (−10.93) | | | | |
| Ln (N_DS_G_S) | | | −0.0110 *** | −0.0650 *** | | |
| | | | (−3.742) | (−16.86) | | |
| Ln (N_DS_B_S) | | | | | 0.0124 *** | −0.00361 |
| | | | | | (3.063) | (−0.859) |
| Constant | 20.29 *** | 2.804 ** | 20.00 *** | 4.159 *** | 19.87 *** | 3.575 ** |
| | (25.71) | (1.966) | (25.31) | (2.975) | (24.83) | (2.474) |
| Observations | 5279 | 4167 | 5279 | 4167 | 5279 | 4167 |
| R-squared | 0.897 | 0.807 | 0.897 | 0.814 | 0.897 | 0.802 |

t-statistics in parentheses; *** $p < 0.01$, ** $p < 0.05$; * $p < 0.1$.

**Table A3.** HPM results are controlled by being close to a park for Measure 2.

| Variable | G1 (1) | G0 (2) | B1 (3) | B0 (4) |
|---|---|---|---|---|
| Ln (living area) | 0.724 *** | 0.732 *** | 0.736 *** | 0.729 *** |
| | (77.13) | (77.70) | (78.19) | (76.83) |
| Ln (number of rooms) | 0.0821 *** | 0.0786 *** | 0.0761 *** | 0.0825 *** |
| | (10.98) | (10.47) | (10.14) | (10.92) |
| Ln (monthly fee) | −0.0990 *** | −0.101 *** | −0.101 *** | −0.104 *** |
| | (−16.59) | (−16.96) | (−16.97) | (−17.21) |
| Ln (build year) | 0.112 | 0.201 ** | 0.0828 | 0.0728 |
| | (1.114) | (1.987) | (0.825) | (0.718) |
| Ln (apartment floor) | 0.0410 *** | 0.0398 *** | 0.0405 *** | 0.0402 *** |
| | (17.59) | (17.01) | (17.37) | (17.07) |
| New | 0.180 *** | 0.181 *** | 0.182 *** | 0.187 *** |
| | (14.92) | (14.95) | (15.05) | (15.36) |
| Elevator | 0.00595 * | 0.00432 | 0.00556 | 0.00362 |
| | (1.738) | (1.258) | (1.621) | (1.048) |
| Ln (dist_subway) | −0.0117 *** | −0.0136 *** | −0.0168 *** | −0.0184 *** |
| | (−4.989) | (−5.763) | (−7.351) | (−7.781) |
| Ln (dist_shop) | −0.00658 ** | −0.00339 | −0.00239 | 0.00475 |
| | (−1.982) | (−1.025) | (−0.732) | (1.458) |
| Ln (dist_CBD) | −0.220 *** | −0.229 *** | −0.235 *** | −0.247 *** |
| | (−29.79) | (−31.16) | (−32.39) | (−33.51) |
| Ln (N_DS_G1_S) | −0.0427 *** | | | |
| | (−14.38) | | | |
| Ln (N_DS_G0_S) | | −0.0313 *** | | |
| | | (−11.87) | | |
| Ln (N_DS_B1_S) | | | 0.0394 *** | |
| | | | (13.72) | |
| Ln (N_DS_B0_S) | | | | −0.0111 *** |
| | | | | (−3.306) |
| Constant | 12.53 *** | 11.75 *** | 12.11 *** | 12.65 *** |
| | (16.76) | (15.61) | (16.17) | (16.70) |
| Observations | 9446 | 9446 | 9446 | 9446 |
| R-squared | 0.873 | 0.872 | 0.873 | 0.871 |

Note: 1—t-statistics in parentheses *** $p < 0.01$, ** $p < 0.05$; * $p < 0.1$; 2—G1: high-quality school close to park, G0: high-quality school not close to park, B1: low-quality school close to park, B0: low-quality school not close to park.

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
