# Peer review of "Schools’ Capitalization into Housing Values in a Context of Free School Choices"

_buildings, doi:10.3390/buildings12071021_

Round 1
Reviewer 1 Report
Thank you for providing me this opportunity to review the paper "Schools' capitalization into housing values in a context of free school choices."
This paper investigates a very important and interesting topic in the literature on housing: the capitalization of proximity in the housing value by testing a unique context of free school choices in Sweden, ruling out the impact of the school zone and boundary effect. After reviewing this paper, I suggest that the journal publish this paper, and I have following suggestions for the authors to consider.
Firstly, in empirical testing, I understand that the author treats the 2019-year data as cross-sectional data, but ideally, the monthly fixed effect should be included to capture the change in housing prices within the year.
Also, according to Figures 1 and 2, the whole of Stockholm is subdivided into suburbs/districts. Therefore, the suburbs/districts fixed effect should be included in the empirical model, and the standard error should be clustered on the suburbs/districts level.
Further, the author may consider utilizing polynomial regression plotting to describe the relationship between housing price and distances to high-ranked and low-ranked schools to validate the findings further. Please consider the polynomial plotting, fixed effect, and standard error clustering model settings of the two following papers:
Diao, M., Leonard, D., & Sing, T. F. (2017). Spatial-difference-in-differences models for impact of new mass rapid transit line on private housing values. Regional Science and Urban Economics, 67, 64-77.
Koo, K. M., & Liang, J. (2021). The effect of bilingual education on housing price-A case study of bilingual school conversion. The Journal of Real Estate Finance and Economics, 62(4), 629-664.
Further, I am concerned that the results may be biased by the endogeneity issue that some low-ranked schools are more likely to be located in the areas mainly resided by refugees, new immigrants, and low-income people. Thus, the distance to the school may capture the local area's demographic profile and socioeconomic features. Some instrumental variables here may address this issue.
Finally, the English language and style are fine, but a minor spell check is required. For example, in line 37, "free choice" should be "free-choice."
Author Response
Review Report #1
Thank you for very good comments! We have adressed all of them.
1- Firstly, in empirical testing, I understand that the author treats the 2019-year data as cross-sectional data, but ideally, the monthly fixed effect should be included to capture the change in housing prices within the year.
- We have included the monthly fixed effect to capture the change in housing prices within the year. The tables and results have been updated, and the new results are approximately the same as our previous results.
2- Also, according to Figures 1 and 2, the whole of Stockholm is subdivided into suburbs/districts. Therefore, the suburbs/districts fixed effect should be included in the empirical model, and the standard error should be clustered on the suburbs/districts level.
- We have controlled spatial location by including zip codes and coordinates in the models.
3- Further, the author may consider utilizing polynomial regression plotting to describe the relationship between housing price and distances to high-ranked and low-ranked schools to validate the findings further. Please consider the polynomial plotting, fixed effect, and standard error clustering model settings of the two following papers:
- We have used polynomial plotting and added it to our paper.
4- Further, I am concerned that the results may be biased by the endogeneity issue that some low-ranked schools are more likely to be in the areas mainly resided by refugees, new immigrants, and low-income people. Thus, the distance to the school may capture the local area's demographic profile and socioeconomic features. Some instrumental variables here may address this issue.
- We used the instrumental variable and added it to the text.
5- Finally, the English language and style are fine, but a minor spell check is required. For example, in line 37, "free choice" should be "free-choice."
- Thanks, we have adjusted them.
Reviewer 2 Report
please see my report

Author Response
Review Report #2
Thank you for very good comments.
1- The manuscript seems to focus on the empirical literature. There is a theoretical counterpart of the school finance, or in general, local public finance literature. For instance, Gong et al. (2020) and Leung et al. (2012) demonstrate that housing policies, whether in the form of public housing or housing subsidies, can impact the spatial sorting of households, which in turn affect the school quality and house prices. Similarly, private schools would also alter the spatial distribution of heterogeneous households and hence the house price (Hanushek et al., 2011). Hanushek et al. (2022) review the literature. The revised edition should relate to that literature.
- Thanks, we have included all the references you mentioned in our paper.
New paragraph included: “Based on a simple spatial equilibrium model, Gong and Leung (2020) analyse the implications of education and housing policies, in terms of public housing, housing voucher programs, and school finance consolidation in the United States. The results of policy analyses indicate that public housing policy can induce a high sorting among the households, which in turn affects the school quality and house prices. Even though, that the housing voucher program can raise the overall welfare of low-income households, it is the parents who capture the welfare gains more than the children, while the public housing program delivers the opposite result. The well-being of children from low-income households could improve by combining the school finance consolidation policy with the public housing program without hurting other households’ welfare [3]. In the same context, Leung et al., (2012) refer that the location of public housing units, the number and sizes of units are fundamental policy variables. They indicate the role of public housing and vouchers in improving overall welfare [4]. Hanushek et al., (2011) indicate that private schools provide households freedom to make residential location decisions in relation to the tax-school quality bundles they consume, which alters the spatial distribution of households and subsequently house prices. Furthermore, private schools contribute to reducing housing and school segregation [5].”
2- A characteristic of the U.S. system is that the local public goods, such as K12 education, are financed by local property tax. Hence, financial resources across different school districts can vary, and teacher salaries across schools can differ (Hanushek et al., 2007; Rivkin et al., 2005). In Sweden, are the salary of teachers identical across schools? Are the financial resources equal across schools? Or is it merit-based? If the teacher salaries are the same and resources are comparable across schools, the quality difference across schools in Sweden may be smaller than that in the U.S. The authors should clarify this.
- We have clarified the questions and added that to the text.
New paragraph included: “Municipalities finance the education system by locally collected tax revenues and state grants. There is no fixed salary scale for schoolteachers, the teachers' salaries are set individually based on position, skills, qualifications, experience, and performance. Sweden's municipalities have vastly different rates of certified teachers with a high gap between municipalities. The disadvantaged suburbs of major Sweden cities, especially Stockholm, have difficulty attracting skilled teachers to jobs in vulnerable areas, which has an impact on the pupils' outcomes [53–55].””
3- The manuscript is also related to a long debate on the determinants of school quality (Hanushek, 2003). If the teacher salaries are the same and resources are comparable across schools, what drives the quality differences across schools in Sweden? Is it reputation, alumni network, or location? For instance, some schools may be located near the waterfront or some historical sites, while some schools are near some less attractive districts. It may affect where the good teachers want to work and the school quality. The same amenity difference across school neighborhoods could also affect the house prices across districts. In other words, the school quality and house price differences could be driven by some third factor, such as amenities. How should we identify the effect of capitalization in this case?
- We have used a set of methods to deal with this case, and we discussed them in the Methodology section.
4- The quality measures need more clarifications.
Some issues need to be clarified. Are these merits value objectively measured? Are they comparable across schools? In other words, can some schools artificially inflate their students' grades?
- Yes, the measure is objectively measured and comparable across schools. We have added that to the text.
An alternative would be the following. If there is a Swedish version of SAT, a university entrance exam, we can measure high school quality. And then, we can rate primary schools according to the percentage of primary school graduates enrolling in top-quality high schools. Of course, it would demand a lot more data and may well be a topic for future research.
- Yes, thanks. It will be a good topic for future research.
Round 2
Reviewer 2 Report
nil